# Droplet Spreading on Unidirectional Fiber Beds

**Patricio Martinez** [1],*, **Bo Cheng Jin** [1,2] **and Steven Nutt** [1]

1   M.C. Gill Composites Center, Department of Chemical Engineering and Materials Science, University of Southern California, 3651 Watt Way, Los Angeles, CA 90089, USA; bochengj@usc.edu (B.C.J.); nutt@usc.edu (S.N.)

2   Advanced Composites Simulation Lab, Department of Aerospace and Mechanical Engineering, University of Southern California, 3737 Watt Way, Los Angeles, CA 90089, USA

*   Correspondence: mart136@usc.edu

**Abstract:** This study reports a method to analyze parametric effects on the spread flow kinetics of fluid droplets on unidirectional fiber beds. The investigation was undertaken in order to guide the design of droplet arrays for production of an out-of-autoclave (OoA) prepreg featuring discontinuous resin distribution, referred to here as semi-preg. Volume-controlled droplets of a resin facsimile fluid were deposited on carbon fiber beds and the flow behavior was recorded. The time to full sorption (after deposition) and the maximum droplet spread distance were measured. Experiments revealed that fluid viscosity dominated time to full sorption—doubling the viscosity resulted in an 8- to 20-fold increase in sorption time, whereas doubling fabric areal weight increased the time only by a factor of three. Droplet spread distance was nearly invariant with fiber bed architecture and fluid viscosity. A series of droplet arrays were designed, demonstrating how the results can be leveraged to achieve different resin distributions to produce semi-preg optimized for OoA cure.

**Keywords:** carbon fibers; prepreg processing; fluid flow; viscosity; wettability

## 1. Introduction

We investigate the effects of fiber bed architecture on the anisotropic flow behavior of fluid droplets on and into unidirectional (UD) fiber beds. In particular, we determine the effects of fiber bed areal weight and fluid viscosity on sorption time and spread distance. The work is motivated by a need to support the design of prepreg formats with discontinuous resin distributions (semi-pregs).

Conventional out-of-autoclave (OoA) prepregs typically feature continuous resin films partially impregnated into the fiber bed and thus rely on "edge breathing" for air removal [1–3]. Compared to autoclave prepregs, however, processing of OoA prepregs lacks robustness, particularly in challenging conditions, such as poor vacuum, ply ramps, embedded doublers and large parts [4,5]. In contrast, semi-pregs [6] feature discontinuous resin distributions that impart high through-thickness permeability and increase process robustness compared to conventional OoA prepregs [5,7–9]. Previous methods of fabricating semi-preg include hot-rolling resin onto the tow overlaps of woven fiber beds [10], using a release film mask to press a discontinuous film onto dry fibers [8] and dewetting a continuous resin film, then pressing onto dry fibers [7].

Recent work on semi-pregs has employed a polymer film dewetting approach to fabrication [7,9,11,12]. The inherent versatility of the method permits deposition of a variety of patterns on the fabric surface. However, the method has limitations. For example, the initial degree of impregnation (DoI) is negligible, leading to a higher bulk factor than conventional vacuum bag-only (VBO) prepreg [12]. In addition, the method as described requires filming of a continuous resin film prior to dewetting and pressing onto the fiber bed, adding cost to the manufacture. Finally, for thin resin films, difficulties can arise in generating uniform discontinuous resin distributions [12]. Thus, alternative methods of

production are being evaluated, including gravure printing and droplet deposition onto fiber beds [13]. Out of these methods, droplet deposition allows for a resin distribution that is controlled based on the relationship between droplet position and fiber bed architecture and does not require a prior filming step as the dewetting method does. The present work constitutes a first step towards semi-preg production by droplet deposition. Experiments were undertaken to understand the flow of a single droplet placed on a fiber bed surface, with further scaling-up used to inform the future design of semi-preg with robust process characteristics.

Fluid flow on porous surfaces and through fiber beds has broad relevance for composites manufacturing [14,15] and has thus been studied extensively. However, most prior studies of surface flow have assumed material isotropy, treating pores as tubes oriented normal to the surface [16]. Fiber beds in composites, however, are anisotropic and permeability varies by orders of magnitude in directions normal and parallel to the fibers [17]. Thus, studies of fluid flow through fiber beds generally must consider flow through an anisotropic porous medium. Most often, such studies consider flow through the entire fiber bed, leading to full saturation (e.g., modeling the impregnation of an individual fiber tow [18]). These studies focus on infiltration, in which one fluid (air) is fully displaced by another (resin) in a porous medium [19], and as such ignore flow above and near the surface of the porous substrate.

Studies of fluid flow on a porous surface generally assume low-viscosity fluids (under 1Pa·s) [20]. Although, during cure, similar low viscosity values are achieved by typical prepreg resins, these studies have limited relevance with regard to surface flow during droplet deposition. The reason for this is that, to prevent advancing cure of the resin, droplet deposition is performed at lower temperatures than those used during cure, leading to higher viscosities. The present study considers higher viscosity fluids and focuses on local (near-surface) flow beneath individual droplets during initial wet-out. The droplets used in semi-preg production must only wet-out partially during the deposition stage in order to preserve connectivity of the dry spaces for air evacuation during the de-bulking stage of VBO processing. Full flow and subsequent saturation of the fiber bed only occurs after de-bulking, during cure of the laminate.

A combination of forces governs the fluid flow of a droplet on a solid surface, including gravitational forces, viscous forces and surface tension [20]. However, the effects of gravity can be ignored for small droplets, leaving only viscous forces, surface tension and capillary forces to govern the flow [21]. Similar forces control the flow of droplets dispersed on the surface of a porous medium saturated with the same fluid [22]. However, for dry fiber beds, capillary effects play a major role [14,15]. Specifically, standard wet-out phenomena lead to droplet spread on the surface, increasing the coverage, while capillary effects promote absorption into the substrate, reducing surface coverage and increasing impregnation [23].

In this study, we measured the surface flow of individual droplets on unidirectional fiber beds. Facsimile fluids with moderate viscosity values were selected (30–60 Pa·s) to resemble polymer resins used during hot-melt production of prepreg. Using the measured response of a single droplet, we also produced droplet arrays to maximize surface coverage, or to minimize interactions between neighboring droplets. The results relate to and can inform the design and production of semi-pregs, particularly the spacings and patterns of resin droplets on fiber beds. Note that despite the match in fluid viscosities, the facsimile fluid did not match the apparent contact angle nor the advancing droplet edge slope. Thus, while the results presented here are useful for determining droplet spread parameters and foreseeing how such droplets will behave in manufacturing conditions, further tests must be performed with actual resin to determine parameters for prepregging.

The experiments revealed parametric effects on droplet spread rates. The droplet absorption time depended strongly on viscosity: doubling the viscosity resulted in an 8- to 20-fold increase in absorption time. However, droplet spread along the surface showed little variation, at least within the fluid viscosity range tested. Similar relationships were noted with fiber bed areal weight, which had little effect on spread distances, but caused

marked changes in sorption time. Finally, surface spread depended strongly on fiber bed architecture, particularly tow gaps caused by stitching. Using these results, we demonstrated how droplets can be positioned on fiber beds to ensure uniform impregnation, informing future methods for semi-preg production.

## 2. Materials and Methods

### 2.1. Resin Characterization and Fluid Selection

An epoxy resin designed for aerospace applications was selected (PMT-F4A, Patz Materials & Technologies, Benicia, CA, USA). The resin viscosity was typical of B-staged resins used in the production of conventional OoA prepreg. During prepreg production, the resin is pre-melted at 65–68 °C (150–155° F), filmed at 68–72 °C (155–162° F), then transferred to the fiber bed. The process from pre-melting to cooling takes less than 90 min. Thus, in the present work, rheological measurements were performed after each step in the thermal cycle, shown in Figure 1a. During the initial pre-melting stage, resin viscosity averaged 49.7 Pa·s, while during the filming stage, resin viscosity averaged 32.4 Pa·s, with a minimum of 27 Pa·s and a maximum of 52.5 Pa·s during the entire cycle.

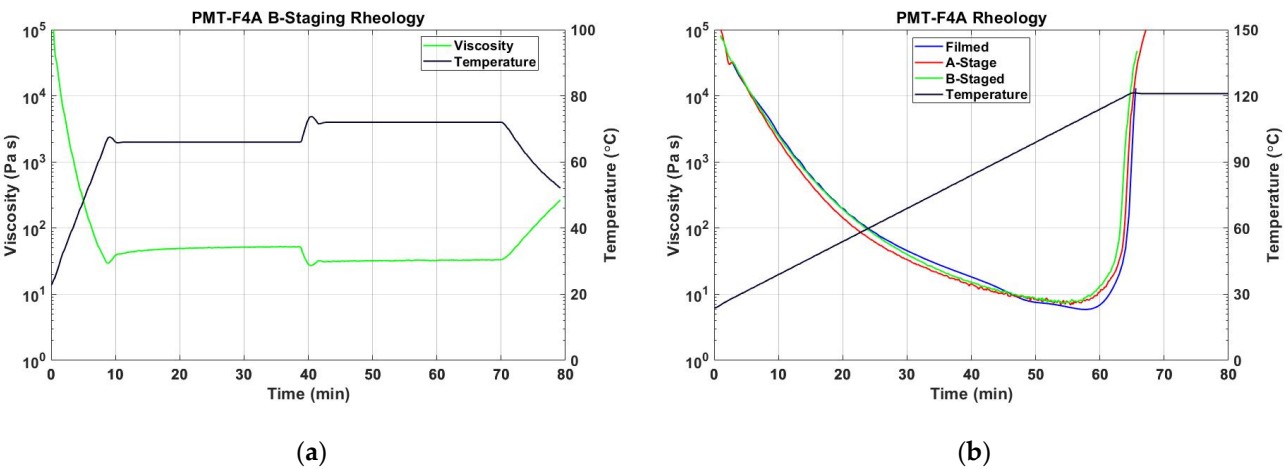

(**a**)  (**b**)

**Figure 1.** (**a**) Rheological profile for PMT-F4A following the filming process thermal cycle. (**b**) Rheological profiles for three separate samples of PMT-F4A.

To ensure that the process was consistent with the manufacturer's practices, the cure cycle rheology was compared with a sample provided by the manufacturer that had not undergone the melting process used for filming (A stage). The pre-melting process showed nearly identical cure cycle rheology (Figure 1b).

Facsimile fluids were chosen to match the viscosity values of the resin during the filming process (Figure 1a). Silicone viscosity standard fluids (General Purpose Silicone, Brookfield Ametek, Middleborough, MA, USA), with viscosities of 30 and 60 Pa·s (±1%), were selected. The use of facsimile fluids enables droplet flow testing at room temperature, minimizing difficulties in maintaining a uniform high temperature on the fiber bed while simultaneously recording data and ensuring the viscosity does not change due to advancing cure.

### 2.2. Contact Angle Comparisons

Surface tension can influence fluid flow along the surface of a substrate and facsimile fluids, despite matching resin viscosity, might exhibit different surface flow behavior. Consequently, measurements of the apparent contact angle were performed, comparing the facsimile fluids to the B-staged resin at the same viscosity. These measurements were performed using a goniometer (Ramé-Hart Model 500, Plymouth, MI, USA) from droplet deposition until apparent stabilization of the contact angle. By determining the difference in contact angle between the facsimile and the resin, a determination can be made of the validity of using the flow behavior observed for a facsimile fluid to predict that of an actual resin.

### 2.3. Fiber Beds

Four unidirectional non-crimp carbon fabrics (UD NCFs) with different areal weights were selected for use as dry fiber beds, including (A) 146gsm (4.3oz/sq.yard), (B) 136gsm (4.0oz/sq.yard), (C) 305gsm (9.0 oz/sq.yard) and (D) 756gsm (22.3oz/sq.yard) (FibreGlast Products #2596, #2585, #2583 and #2595, respectively). UD NCFs were chosen because of previous work with similar fabrics [6–8], their simplified geometry compared to woven fabrics, the similarity to tapes used in automated tape layups and their growing use in vacuum infusion processes both in aerospace and wind blades. Carbon fiber fabrics were selected due to their common use in prepreg for the aerospace industry. Using a different fiber material would result in different intra-tow capillary sizes based on fiber diameter, as well as differences in wettability based on fabric-fluid surface parameters, leading to differences in flow. The fabrics contained polyester binding to stitch layers together and impart ease of handling. The first three NCFs were bound using polyester stitching perpendicular to the fibers on one side, spaced ~10 mm apart. The heaviest weight fabric, however, featured binding in a diamond pattern and fiber tows with distinct edges, as shown in Figure 2.

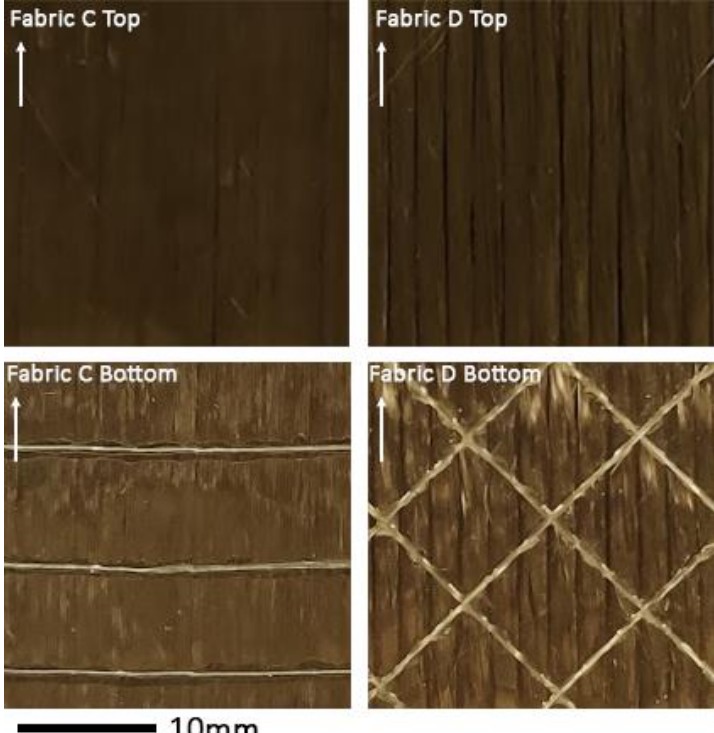

**Figure 2.** Top and bottom views of the two distinct types of fabrics. Fiber direction is indicated with arrows.

Prior to droplet deposition, a 19 mm strip of fabric was cut and the edges were secured with tape to prevent fraying, leaving a 19 × 19 mm square of exposed fabric. The tape was positioned perpendicular to the fiber direction, preventing free fiber edges from lifting and distorting the surface. The fiber bed squares were examined using a digital stereo microscope (Keyence VHX-5000, Osaka, Japan) to generate images and 3D contour maps of the surface. The fabric properties are summarized in Table 1.

**Table 1.** Comparison of properties for the different fiber beds. NCF, non-crimp carbon fabric.

| Fabric | Areal Weight [g/m$^2$] | Tow Count | Fabric Style |
| --- | --- | --- | --- |
| A | 136 | 12 K | Standard NCF |
| B | 146 | 12 K | Standard NCF |
| C | 305 | 24 K | Standard NCF |
| D | 768 | 24 K | Quilted NCF |

### 2.4. Droplet Deposition

Two devices were used to monitor the droplet deposition experiments: a goniometer and a camera (LUMIX GH4, Matsushita Electric Co., Osaka, Japan). The sample was positioned on the goniometer stage, with fibers aligned perpendicular to the goniometer light source and camera. Samples were positioned with stitching facing downward. The second camera was positioned directly above the sample (Figure 3). A syringe was used to deposit a single droplet of the silicone oil facsimile and both cameras started recording as the droplet contacted the fiber. After droplet deposition, the syringe was removed from the field of view.

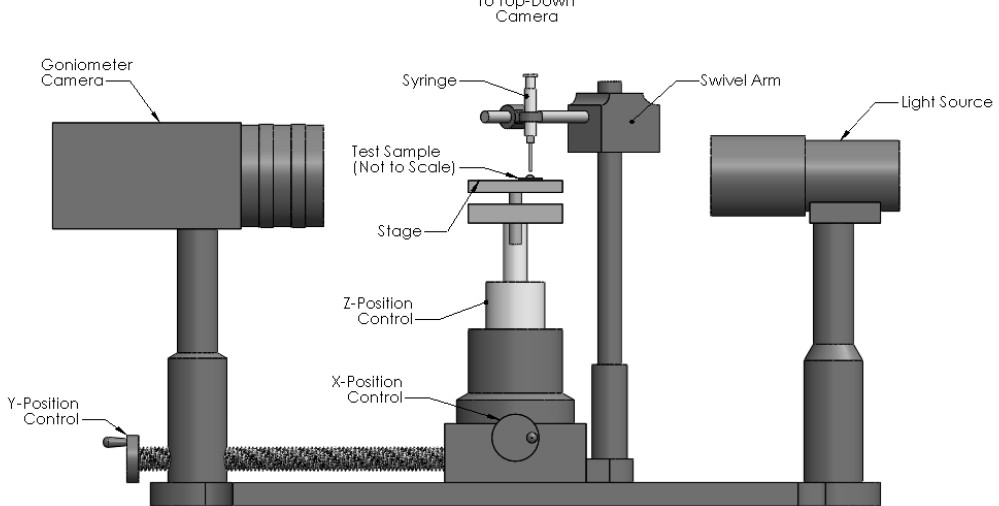

**Figure 3.** Diagram showing the droplet deposition test set up.

The goniometer camera was used to record images at two-second intervals, while the top-down camera recorded images at ten-second intervals. The images were compiled and analyzed separately. Software (MATLAB R2019a) was used to analyze the goniometer images, while the top-down images were analyzed manually using image editing software. Droplet perimeter was approximated using a brightness threshold, then further modified manually, as the contrast between wet and dry was not perfect. In both cases, the dimensions of the fluid spread were measured from images. Using the side view, droplet width and height were measured at each frame. Using the top view, the droplet spread was measured in directions parallel and transverse to the fibers. Figure 4 shows diagrams illustrating the measurements recorded, as well as an example of one such frame used for a single measurement. The goniometer camera detected fluid above the fiber substrate only, while the top-down camera allowed for the observation of fluid imbibed by the fiber bed.

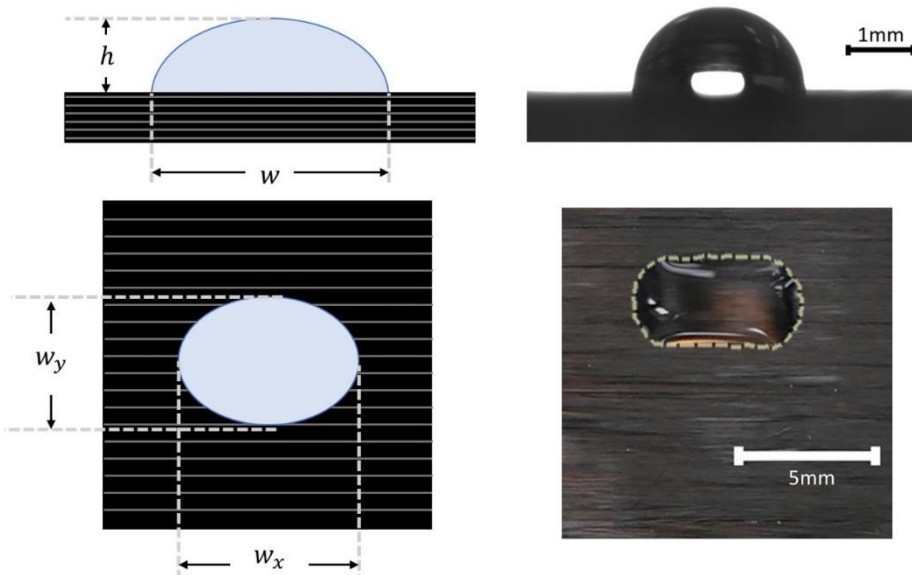

**Figure 4.** Diagram showing the measurements made from side and top-views of the droplet deposition test (**left**) and an examples of images used for such measurements (**right**), with the droplet edges outlined for clarity in the top-view.

### 2.5. Droplet Grid Tests

Droplet arrays were deposited to demonstrate how the results from previous sections can inform designs of resin patterns on semi-pregs. Droplet arrays of the facsimile fluid (30 Pa·s) were deposited on a 20 × 20 mm area of fabric A. The arrays were evaluated with respect to two parameters—(a) area covered by the fluid and (b) neighboring droplet interaction. Three distinct droplet arrays were deposited. The first pattern consisted of droplets uniformly distributed in a 3 × 3 square grid. The second pattern was based on data obtained from the single droplet tests, allowing the droplet positions to be arranged such that droplet overlap was minimized. The final grid pattern used the same data with minor changes to the positioning to ensure droplet-to-droplet interactions, showing how small deviations can result in differences in the final distribution.

Grid accuracy was maintained by producing a guide for the same syringe used for the single droplet deposition tests. Droplets were aligned using the grid guides and dispensed one at a time in raster fashion on the fiber bed. Droplet spreading was recorded using a top-down camera in the same manner as for the single droplet tests. Since the positioning guide shielded the camera view, data recording commenced 5 min after the first droplet was deposited. Images were captured at 10-sec intervals for up to 150 min. Using these images, the area covered by facsimile fluid was recorded over time. Furthermore, a time lapse was generated using the captured images for each droplet array.

### 3. Results

#### 3.1. Contact Angle Comparisons

Using all three fluids—the resin and the two different viscosity silicone oils—differences in surface flow phenomena were observed and recorded. Experiments were conducted to measure the apparent contact angle as the droplet was deposited and absorbed into the fabric. As shown in Figure 5a, the facsimile fluid did not match the apparent contact angle of the resin. Given the non-static nature of the measured angle, it is more appropriate to refer to the metric not as the contact angle, but as the edge slope of the droplet. However, as shown in Figure 5b, the difference in edge slope between the lower and higher viscosity droplets was equivalent for both the epoxy resin and the facsimile fluid.

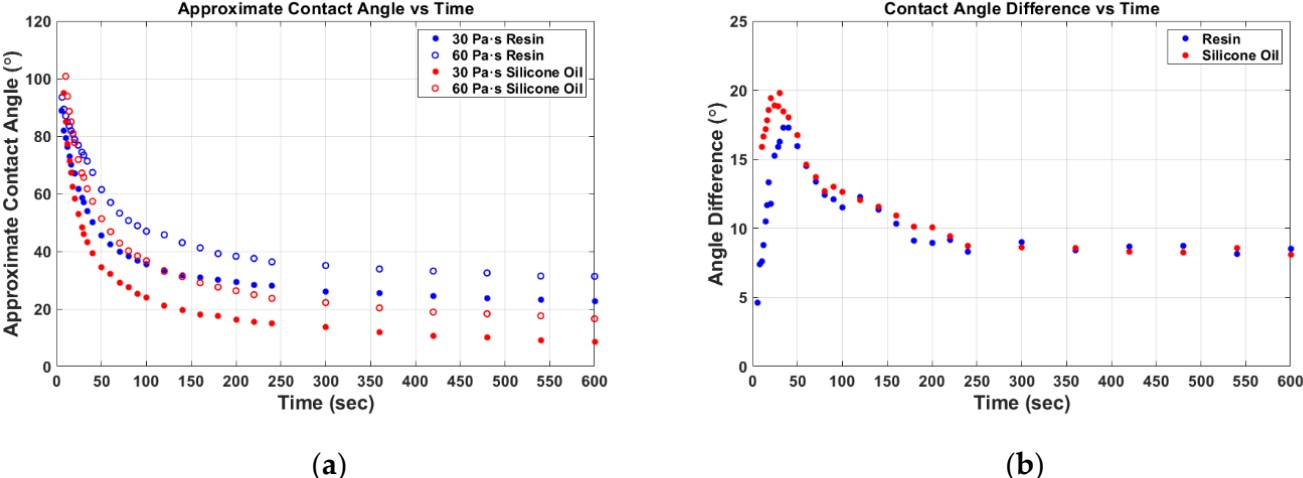

**Figure 5.** (**a**) Apparent contact angle of droplets of silicone oil and epoxy resin at 30 and 60 Pa·s viscosity. (**b**) Difference between the apparent contact angle of the 30 and 60 Pa·s droplets for both resin and facsimile fluid.

### 3.2. Single Droplet Test

Experiments were performed for each substrate–facsimile fluid combination, while recording droplet height ($h$), droplet width above the surface ($w$), fluid spread across fibers ($w_y$) and fluid spread along fibers ($w_x$). Note that $w$ and $h$ were recorded from the goniometer, while $w_x$ and $w_y$ were recorded from the top-down view. While both $w$ and $w_x$ represent droplet dimensions in the direction parallel to the fibers, $w$ represents the width of the droplet seen above the surface, while $w_x$ includes fluid flow visible at the surface from the top-down view. The time to full sorption ($t_{h0}$) was taken as the time required for the droplet height to reach a constant value. This time was used to normalize the remainder of the time for the figure, as follows:

$$\hat{t} = \frac{t}{t_{h0}}$$

Figure 6 shows (a) droplet height and width versus time and (b) test results normalized by time to full sorption. When plotted against the logarithm of normalized time, the height of the droplet decreased approximately linearly. Similarly, the spread distance along the fibers also increased linearly, which follows from Tanner's Law (1) for a two-dimensional droplet, where $R(t)$ is the radius of the droplet, $\gamma$ is the surface tension, $B$ is a constant, $\eta$ is fluid viscosity and $V$ is droplet volume [24,25]. In summary, the spread distance along the fibers, $w_x$ in this case, follows a power law with time. In contrast, the spreading behavior across the fibers was distinctly nonlinear (albeit noisy), exhibiting spread at the start followed by a quasi-stable plateau. Droplet spreading generally followed patterns similar to those depicted in Figure 6, with variations in the rate of growth and decay.

$$R(t) \approx \left[ \frac{10\gamma}{9B\eta} \left( \frac{4V}{\pi} \right)^3 \right]^{\frac{1}{10}} \propto t^n, \tag{1}$$

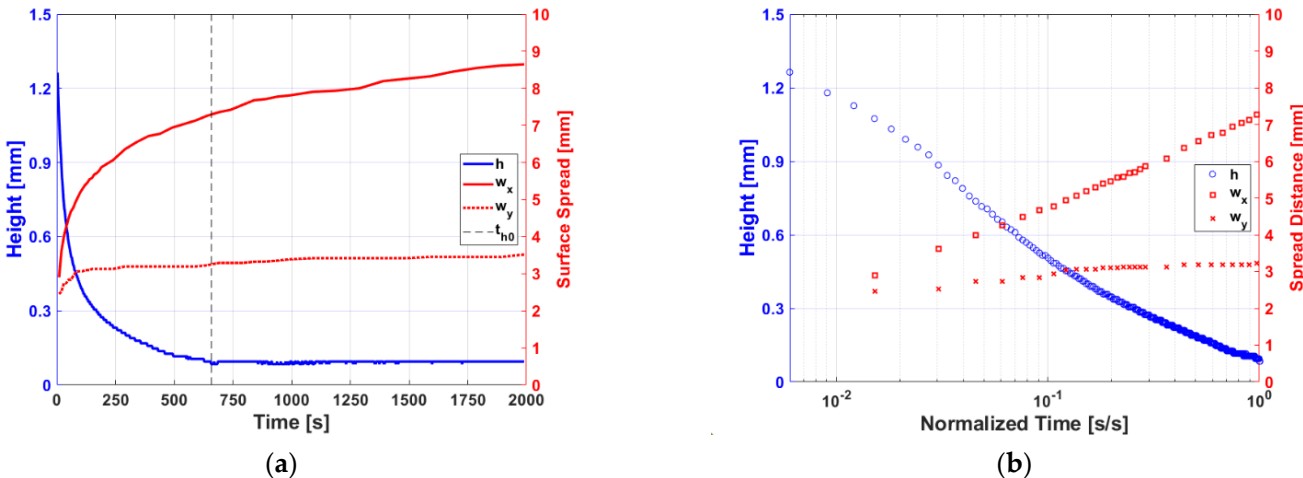

**Figure 6.** (**a**) Results from depositing 30 Pa·s viscosity fluid on fabric C. The left *y*-axis shows results derived from the side view, while the right *y*-axis shows results from top-down view. (**b**) Same results as (**a**), with time scale normalized to $t_{h0}$.

### 3.3. Aggregate Tests Results

To visualize the evolution of droplet height with time, select images were assembled in an array, as shown in Figure 7. For these images, the height of the droplet was mapped against fractions of time to full sorption, $t_{h0}$. These images show that as fabric areal weight increased, in-plane spreading of droplets generally occurred more rapidly, the only exception being the fabric with the heaviest areal weight. In the case of the standard NCFs with the highest viscosity droplets, most of the spreading occurred early; that is, the fluid spread out rapidly before it started being absorbed into the fabric. The effects of gravity in assisting flow were strongest in early stages, when droplet mass was most centralized.

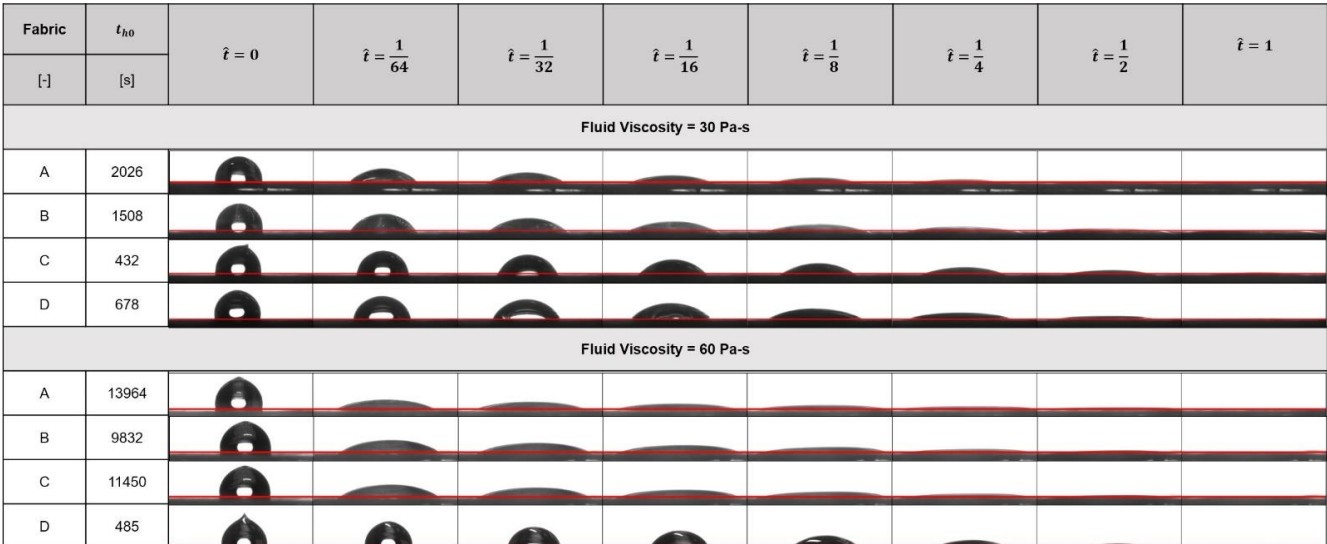

**Figure 7.** Time lapse of side view for all combinations of fluid viscosity and fabric types. Each snapshot is at a fraction of the time to full sorption for that test. Red line corresponds to the baseline from which height is measured, equal to the height of the droplet at $t = t_{h0}$ ($h = 0$).

Figure 8 was generated using the time to full sorption, $t_{h0}$, for each test. For the first three fabric samples, lower areal weight correlated with increased time to full sorption for the low-viscosity oil. When fluid viscosity increased (from 30 to 60 Pa·s), $t_{h0}$ markedly increased (between 8- and 20-fold). In contrast, no similar correlation appeared for the heaviest fabric: fluid viscosity did not affect time to full sorption for the 756 gsm fabric.

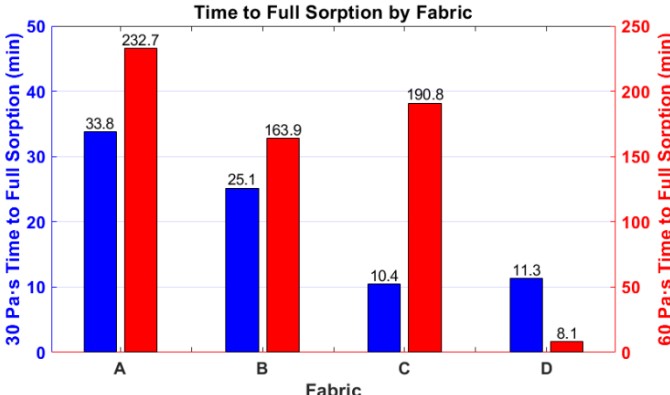

**Figure 8.** Relationship between time to full sorption and fabric type. Note that the 60 Pa·s samples have been cut off, but the relative height between each of the 60 Pa·s tests (excluding fabric D) remains.

Figure 9 compares the maximum fluid flow distance in both directions of interest for each test. As expected, the fluids spread longer distances along the fibers than across the fibers by a factor of 2–3×. For the first three fabrics, spread distances along the fibers fell within a 1.9 mm range and spread distances across the fibers were within a narrower 1.1 mm range. Spread distances along the fibers did not fall in this range for fabric D, at least for the lower viscosity fluid.

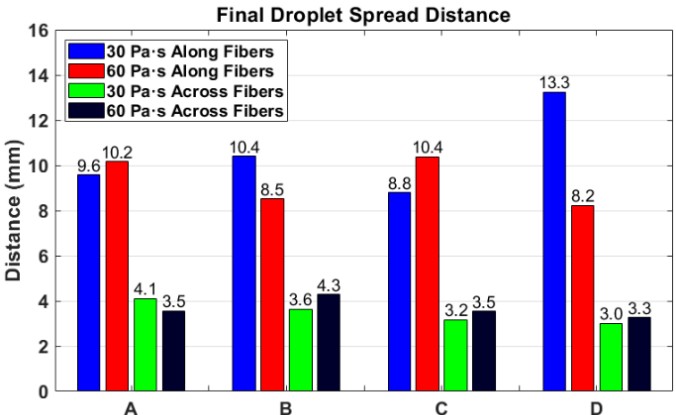

**Figure 9.** Relationship between fiber bed type and spread, in directions across and along the fibers.

### 3.4. Quilted Fiber Bed

Fabric D behaved differently from other fabrics due to its distinct architecture. This fabric featured individual tows secured by a grid pattern of stitches, as opposed to simple unidirectional stitches (shown previously in Figure 2). The gaps between individual tows afforded pathways for fluid flow into the fiber bed, acting effectively as macrochannels between fiber bundles. Top-down images of fabric D revealed that most of the fluid flowed into the gaps (shown overlaid on a topographical image of the fiber bed in Figure 10). The results in the previous section showed that fabric D yielded the largest fluid flow distance along the fibers, accompanied by minimal flow across the fibers. These observations support the assertions that irregularities in flow along the fibers were caused by greater fluid penetration into the surface, and that penetration occurred through and along the larger inter-tow gaps created by stitching. Comparing this result to a similar test on one of the non-quilted fabrics, the bottom image in Figure 10 shows that fluid also flowed into gaps/irregularities within the fiber bed, but that the flow was spread more uniformly along the fibers.

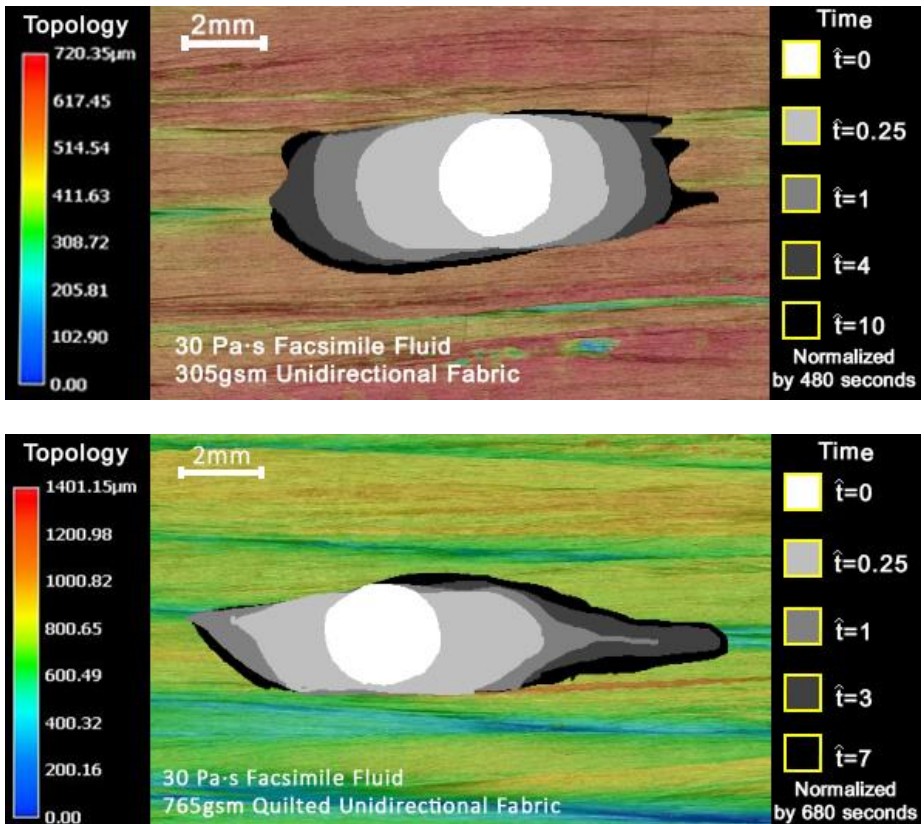

**Figure 10.** Images showing the contours of the area infiltrated by fluid over time, overlaid onto a topographical representation of the fiber bed for the 30 Pa·s viscosity fluid tests on fabric C (**above**) and fabric D (**below**).

The observations in Figure 10 highlight challenges when attempting to correlate the results reported here to other fabric types. Fine-scale variations in the fiber bed, such as inter-tow gaps, pinholes and other such irregularities common in woven fabrics, are intrinsic and affect fluid flow. However, the impact of such variations on the flow of an individual droplet is likely to be more pronounced. Due to the similarity in length scales of droplet size and surface features, droplet flow is dominated by the positioning of the droplet on the surface. For example, in the case of the quilted unidirectional fabric above, a droplet at the center of a tow bundle is likely to behave differently from one deposited directly atop a tow gap, as shown in the top image of Figure 10. Since the non-quilted UD NCFs, fabrics A–C, do not exhibit large-scale surface irregularities, the results from these fabrics can be analyzed with fiber areal weight as the only variable.

*3.5. Droplet Grid Optimization*

Using the results from Figure 9, three droplet arrays were designed, including a control pattern, with droplets arranged in a square grid, a staggered array to maximize coverage and minimize droplet overlap and a tight-staggered array that ensured droplet interaction while maximizing spread distance (grids 1–3, Figure 11). Coverage refers to the area fraction of the surface that was covered by the facsimile resin upon full sorption and overlap refers to neighboring droplets impinging on each other and coalescing into a single fluid pool.

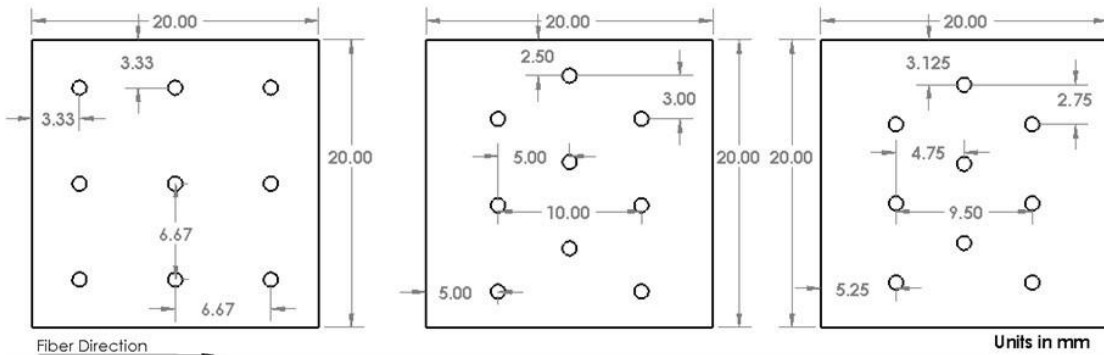

**Figure 11.** Diagrams of droplet grid guides used. From left to right: 1. square grid, 2. staggered grid, 3. tight-staggered grid.

The dimensions of the grid arrays can be used to estimate the fiber loading and resin content for a fully impregnated prepreg. Using the controlled volume of the droplet, the areal weight of fabric A and a fiber density of ~1.7 × 106 g/m³, eighteen droplets on a 20 × 20 mm area correspond to a fiber volume fraction of 60–70%, assuming full impregnation and no voids. This range of fiber loadings lies within the range used for commercial unidirectional prepregs. Since commercial prepregs are produced by applying resin to both sides of a fiber bed, half the droplets needed to achieve a proper volume fraction were used and nine droplets were placed within a 20 × 20 mm area for the droplet grids.

Diagrams of the three grids are shown in Figure 11. Grid 1 featured droplets separated by 6.67 mm in a square array that extended across and along the fibers. The design for grid 2 was informed by the observations that droplets spread 9.6 mm along the fibers and 4.1 mm across them. The droplets were spaced 10 mm apart along the fibers and 3 mm apart across the fibers, with an offset of 5 mm along the fibers between rows. Grid 3 featured a modification of grid 2, with droplets spaced 9.5 mm apart along the fibers and 2.75 mm apart across the fibers.

Figure 12 shows the final images captured, as well as contours for the area covered by the fluid for the first and last images. The images show that all three arrangements resulted in droplet overlap. However, by design, grid 2 exhibited the least overlap, while grid 3 yielded the most. The square grid, grid 1, showed overlap along the fibers and dry gaps between droplets (across the fibers). Green and red outlines in Figure 12 show initial and final perimeters of the nine initial droplets, respectively, as seen from the surface. At the end of each test, the red outlines show that grid 1 resulted in three distinct fluid-covered regions, grid 2 with six and grid 3 with two. Furthermore, upon completion of each test, the portion of the fluid area that extended beyond the prescribed region was 5.3%, 0.6% and 3.2% for grids 1, 2 and 3, respectively. These fluid regions were determined only above the surface and do not include subsurface flow.

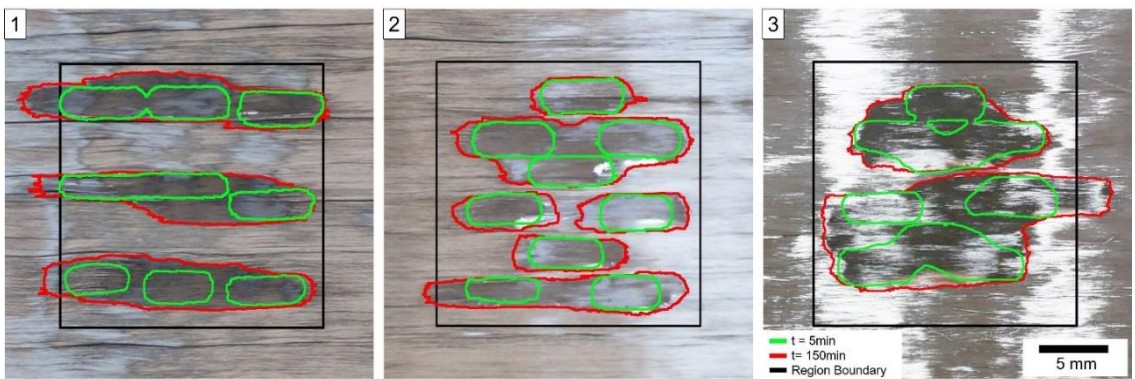

**Figure 12.** Images showing the extent of fluid coverage for all three grids, including 5 min and 150 min after deposition.

The area covered by the droplets was recorded and plotted versus time (Figure 13). This graph shows that grids 2 and 3 covered greater areas than grid 1, but the difference was small (3–4%). Upon completion of the tests, grids 1–3 covered 192.44, 198.05 and 200.48 mm$^2$, respectively.

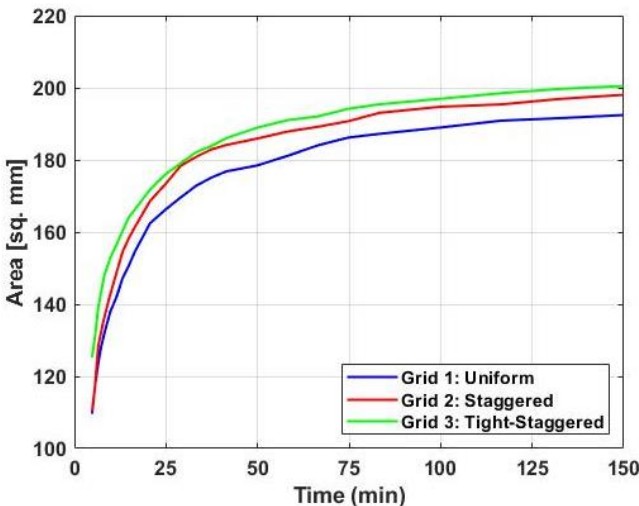

**Figure 13.** Graph showing the area covered by the droplets using all three grids.

## 4. Discussion

### 4.1. Absorption Kinetics

The time to full sorption, $t_{h0}$, was inversely proportional to fiber bed areal weight. As shown in Figure 8, for 30 Pa·s droplets, fabrics A, B and C exhibited $t_{h0}$ values of 33.8, 25.3 and 10.5 min, respectively. Furthermore, only fabric A exhibited full through-thickness penetration of the fluid, as evidenced by the presence of facsimile fluid residue on the underside of the fabric. From these results, we assert that heavier fabrics allowed for deeper through-thickness flow of the droplet, reducing the time-to-saturation for droplets of the same size.

In contrast, increasing fluid viscosity generally increased sorption time, although the relationship between viscosity and $t_{h0}$ was not proportional (based on the two silicone oils tested). Figure 8 shows that, except for fabric D, increasing fluid viscosity by a factor of two led to an 8- to 20-fold increase in sorption time. In fabric D, high-viscosity resin readily penetrated the macrolevel features (features that were absent in fabrics A through C).

The mechanism by which droplet height changed depended on fluid viscosity. For example, the height change of the 60 Pa·s droplets was driven primarily by spreading on the fabric surface, while the height change of 30 Pa·s droplets was dominated by more rapid penetration into the fabric, shown previously in Figure 7. Higher viscosity fluid droplets spread longer distances along the fiber bed surface, in some cases extending beyond the field of view. In contrast, droplets of the lower viscosity fluid remained within the field of view until full sorption was reached. The 60 Pa·s droplets showed reduced absorption and spread to a large area, resulting in extended absorption times.

The most important metric for production of semi-pregs via droplet deposition is the time to full sorption, referring to the time until the droplet is fully within the fiber bed. As such, by ensuring that a deposited droplet can achieve maximum spread within the time to full sorption, it should be possible to produce prepreg with a degree of impregnation high enough to achieve bulk factors similar to standard OoA prepregs (~10%). As shown in Figure 6, droplet flow continued after $t_{h0}$, allowing the droplet to flow further within the fiber bed and ensuring full saturation of the fiber bed during cure. However, in practice, a balance must be achieved between the DoI, bulk factor and droplet separation. In particular, the spreading time must be close enough to $t_{h0}$ to achieve both a high DoI and low bulk factor, while still maintaining droplet separation and allowing for full ply

saturation during cure. Furthermore, given the differences in $t_{h0}$ achieved by the minor increase in viscosity used in this study, droplet spreading can be virtually arrested by simply cooling the prepreg.

### 4.2. Spread Kinetics

Fabrics A–C were all uniform, unidirectional fabrics with similar smooth surfaces, as shown in Figure 2 (left), and as such similar droplet spread behavior was expected for these fabrics. Figure 9 shows that droplet spread distances fell within a narrow range, regardless of fiber areal weight. Furthermore, there was no correlation between spread distance and areal weight. Droplet spread distance was governed solely by surface topography, while areal weight had a negligible effect.

The droplet spread was characterized by two stages as well as two different directions. The two spreading stages were (a) rapid spreading while the droplet remained atop the fabric surface, followed by (b) slower spreading once the droplet was fully imbibed. In the test shown in Figure 6, 80% of spreading along the fibers occurred before full sorption, and 90% of spreading across the fibers occurred before full sorption. Spreading above the surface was driven primarily by gravity and surface tension, while spreading within the fiber bed was driven by capillary effects. The spreading was also different along and across the fiber direction and droplets spread much longer distances in the fiber direction than across fibers. Furthermore, most of the spreading occurred before sorption in the through-thickness direction. The early stage spreading was aided by capillarity between aligned fibers, facilitating fluid flow along fibers and impairing flow across the fibers after full absorption.

For fabrics A–C, the capillary radius was expected to remain the same, and when using the same viscosity fluid, the remaining parameters were similarly unchanged. Therefore, the distance traveled within the capillary, and as such the final distance spread, remained the same, as expected. The horizontal capillary flow of a droplet is described by Washburn's Equation (2), where $L$ is the distance traveled within the capillary, $\gamma$ is the surface tension, $r$ is the capillary pore radius, $t$ is time, $\theta$ refers to the contact angle and $\eta$ is the viscosity [26].

$$L = \sqrt{\frac{\gamma r t \cos(\theta)}{2\eta}}, \tag{2}$$

An increase in viscosity was shown in Figure 5 to result in an increase in contact angle, meaning the change in silicone oil resulted in two terms changing within Washburn's equation. Furthermore, referring to Equation (2), it can be seen that the increase in viscosity and increase in angle effectively counteracted one another, resulting in only minor deviations in spread distance.

### 4.3. Effects of Macrofeatures

Macroscopic features of fabrics, particularly inter-tow gaps, strongly influenced droplet spread. For example, Figure 10 showed that fluid flowed quickly into inter-tow regions of fabric D. These regions served as channels for macroflow, as opposed to microflow within tows. Similar results can be expected for woven fabrics, where the placement of the droplet, particularly with regard to the proximity to pinholes in the weave, affects droplet spread more strongly than fabric areal weight or intra-tow capillary sizes.

With regard to developing a method for predicting droplet spread, an analytical solution or a numerical simulation would be useful for unidirectional fabrics. However, for woven fabrics, which feature complex surface topography, the parameters of the fabric may have only minor effects on surface spread compared to the position of the droplet on the fabric terrain. Generating an analytical solution or simulation for such fabrics would require an added level of complexity and rely on the specific surface features of the given weave.

### 4.4. Droplet Arrays

Droplet positioning and array designs can be guided by the results presented here, as shown by the observations of spread behavior in the three different droplet grids. The findings support the hypothesis that fluid flow occurs more rapidly along the fibers than across and provide approximate indications of how a droplet will spread along the surface. The spread distances can be used to arrange droplets in arrays that prevent/minimize fluid overlap and that ensure maximum, controlled flow distances. However, these same results can also be used to ensure full surface coverage, maximizing overlap and flow distances, as shown in Figures 12 and 13. For example, grid 3 exhibits overlapping fluid covered regions while maintaining similar coverage as the other arrays. Due to the strong degree of anisotropy of unidirectional fiber beds, droplet positioning must be staggered to ensure maximum surface coverage.

The droplet grid tests also revealed effects of droplet impingement. Neighboring droplets influenced the direction of spread and in some cases altered the spreading behavior that would be expected for a single droplet. As shown in Figure 12, once contact between droplets occurred, droplets quickly spread to fill regions between neighboring droplets. Flow occurred even across the fiber bed when driven by the surface tension of the fluid. Transverse flow was responsible for the increase in surface area covered by grids 1 and 2, which included droplet interaction across the fiber bed.

Using these insights into single-droplet behavior, droplets can be positioned in arrays that minimize droplet interaction and maximize through-thickness air evacuation pathways. Alternatively, droplets can be positioned to intentionally create a degree of interaction, achieving a higher surface coverage. These capabilities can be leveraged for use in semi-preg design, which requires discontinuous resin distribution, and more careful control of the resin distribution is achievable with this method than with currently implemented methods. Furthermore, should other novel composite designs emerge that rely on discreet resin droplets, this study may serve as an initial guide.

### 4.5. Facsimile Fluid

Given the impact of droplet–substrate surface parameters on the driving forces behind droplet spreading, matching viscosity is not sufficient to ensure accurate simulation of resin droplets. Experiments showed that the apparent contact angles, or the evolving edge slopes, did not coincide between the facsimile fluid and the resin, and as such the observed results of spreading distance and absorption times for the facsimile fluid cannot be assigned one-to-one to the resin. However, as Figure 5b shows, the relationship between the 30 Pa·s droplet and the 60 Pa·s droplet was maintained both for resin and the silicone oil facsimile. Based on this result, we assert that the relative effects of increasing the viscosity are equivalent when using actual resin droplets.

### 5. Conclusions

We have demonstrated the effects of unidirectional fiber beds on droplet flow on the surface. The fabric feature that most strongly influenced the surface flow was the fiber bed architecture. Macrolevel features of the quilted UD fabric (fabric D) dominated the surface flow of droplets. Decreasing fiber areal weight and increasing viscosity both led to an increase in time to full absorption. However, neither of these factors appreciably influenced the droplet surface coverage.

The findings presented entail multiple implications. First, predicting droplet surface flow is most reliable for unidirectional fiber beds and the anisotropy of UD can be exploited to design patterns that prevent or minimize droplet impingement. By heating or cooling the fluid, viscosity values can be adjusted to allow longer working times for droplet deposition. Within the narrow viscosity range used in this study, the actual droplet spread distances would see little variation. These findings are potentially useful, yet challenges remain. For droplet deposition onto woven fabrics, the specific interactions between droplets and macrolevel details of the fabric must be tracked. In addition, an actual resin

must eventually be used, as opposed to a facsimile fluid. Finally, single droplet size was monitored throughout the study. However, the volume of resin needed to achieve a proper volume fraction for prepreg differs with the fabric weight and smaller droplets are likely more reliable for thinner fabrics. Attending to these challenges will inevitably fall to those designing a droplet deposition system.

The findings highlight the groundwork required to develop practical methods to produce semi-pregs and indicate a pathway to achieving more efficient and robust OoA production. The spread of resin after deposition can be manipulated by determining the time allowed for flow into the fiber bed, as well as droplet position in relation to other droplets and to the fiber bed, allowing for intelligent design of resin distribution in semi-preg. Such designs will facilitate the scaling-up for manufacture of semi-pregs via droplet deposition, as opposed to current methods that rely on prepreg manufacture one ply at a time. Semi-preg is a robust intermediate material for OoA processing and can potentially restore robustness to levels comparable to conventional autoclave manufacturing. In addition, OoA processes reduce costs and increase part throughput, allowing for increased part complexity and accessibility.

**Author Contributions:** Conceptualization, P.M., B.C.J. and S.N.; methodology, P.M.; software, P.M.; validation, P.M. and B.C.J.; formal analysis, P.M.; investigation, P.M.; resources, S.N.; data curation, P.M.; writing—original draft preparation, P.M.; writing—review and editing, P.M., B.C.J. and S.N.; visualization, P.M.; supervision, B.C.J. and S.N.; project administration, P.M. and S.N.; funding acquisition, S.N. All authors have read and agreed to the published version of the manuscript.

**Funding:** This research was funded by the National Science Foundation Partnerships for Innovation, grant number 53-4504-7788.

**Acknowledgments:** The authors thank Mark Anders at the M.C. Gill Composites Center for guidance and support, as well as the M.C. Gill Composites Center for material support.

**Conflicts of Interest:** The authors declare no conflict of interest.

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
