# Peer review of "Droplet Spreading on Unidirectional Fiber Beds"

_jcs, doi:10.3390/jcs5010013_

Round 1

Reviewer 1 Report

This manuscript reports a parametric study of effect of fluid viscosity and fiber bed architectures on the sorption time and spreading distance of fluid droplets on unidirectional dry fiber tows. The work is novel and its findings are relevant, in particular to the out-of-oven prepreg fabrication. The method employed is scientifically sound. The manuscript is well written. Therefore, the review recommends acceptance with minor revision. The reviewers has a few comments for the authors to consider to further improve the quality of the manuscript.

General comments:

  • It is reasonable that facsimile fluids were used in place of actual resins for experimental practicality; however, as the authors also mentioned in the manuscript, for this work to have a larger impact, experiments with real resins should be carried out in the future.
  • Have the authors considered the resin viscosity (and other properties) change during the curing process?
  • Is there any capability to measure penetration depth?
  • Would different fiber types affect the results?

Specific comments:

  • Figure 2. Please consider annotating the fiber directions.
  • Figure 3. Sketch a sample on the sample stage would be a nice addition.
  • Page 5: “…The images were analyzed manually…” Could the authors provide more details on how the images were analyzed?
  • Page 11: Lines 334—343. Are these blue and red boundaries defined manually? If so, how reliable/subjective are the quantifications?

Reviewer 2 Report

Dear Author,

The present work, dealing with droplet spreading on fibers, is quite interesting for representing a step forward towards more efficient and robust OoA composites production. It was particularly pleasing to read a state-of-the-art review that is not exaggerated and yet contains highly relevant data, and to learn that adopting matching viscosity fluids for wettability studies might not be sufficient to ensure accurate simulation of resin droplets. Also, compliments for Figure 7.

In order to recommend the acceptance of your work, I hereby list a few topics to be covered:

  1. Both abstract and introduction promptly start with the objectives of the work. Instead, I advise the addition of a couple of sentences in their beginning that actually introduce the matter in question in terms of “what is the overall topic”, then “what is the challenge in this field that you identified”, and finally “how did you approach this challenge to solve/investigate it”. Otherwise, even a professional from the composites area might not be able to identify the actual purpose of the study. In the introduction section, for instance, it might be enough only to move the current first paragraph to the end of the section.

  1. In Figure 1, the Temperature axis is partially hidden. Please, adjust the size of the figure.

  1. Please state into subsection 2.3 which type of fiber you are dealing with (carbon). Furthermore, it is known that wettability may differ depending on the fiber. Please, explain why have you chosen only carbon.

  1. Figure 6: again, it is not possible to read the title of the axis to the right. The resolution of the images should be improved. Adding, Figures 5, 6, 8 and 9 were formatted poorly, seems like they were just retrieved from Excel with barely no editing. Please improve their visual aspects.
